# The Role of Alpha-Fetoprotein (AFP) in Contemporary Oncology: The Path from a Diagnostic Biomarker to an Anticancer Drug

**DOI:** 10.3390/ijms24032539

**Published:** 2023-01-28

**Authors:** Joanna Głowska-Ciemny, Marcin Szymański, Agata Kuszerska, Zbyszko Malewski, Constantin von Kaisenberg, Rafał Kocyłowski

**Affiliations:** 1PreMediCare New Med Medical Center, ul. Czarna Rola 21, 61-625 Poznań, Poland; 2Department of Perinatology and Gynecology, Poznan University of Medical Sciences, 60-535 Poznań, Poland; 3Department of Obstetrics and Gynecology, Hannover Medical School, Carl-Neuberg-Str. 1, D-30625 Hannover, Germany

**Keywords:** liver cancer, alpha-fetoprotein, AFP, immunotherapy, HCC, GIP, AFPep

## Abstract

This article presents contemporary opinion on the role of alpha-fetoprotein in oncologic diagnostics and treatment. This role stretches far beyond the already known one—that of the biomarker of hepatocellular carcinoma. The turn of the 20th and 21st centuries saw a significant increase in knowledge about the fundamental role of AFP in the neoplastic processes, and in the induction of features of malignance and drug resistance of hepatocellular carcinoma. The impact of AFP on the creation of an immunosuppressive environment for the developing tumor was identified, giving rise to attempts at immunotherapy. The paper presents current and prospective therapies using AFP and its derivatives and the gene therapy options. We directed our attention to both the benefits and risks associated with the use of AFP in oncologic therapy.

## 1. Introduction

Alpha-fetoprotein (AFP) has been known to medicine since the 1960s. Until the 1990s, it was used as a marker for fetal neural tube defects [1,2,3,4] and liver cancer, to become the focal point of interest for molecular biologists later, due to its characteristics as a cell growth modulator in the neoplasm process, which, depending on the conditions, was responsible either for the proliferation or apoptosis of neoplastic cells. At the turn of the 20th and 21st centuries, AFP and its peptide fragments were tested as a therapeutic agent in the treatment of hepatocellular carcinoma (as an anticancer drug carrier and element of immunotherapy vaccines) and investigated as part of research on multiple drug resistance to chemotherapeutics in HCC [5,6,7]. This became the source of multiple benefits, but at the same time multiple threats, as even today, after years of research, we have still failed to discover all the biological properties of this molecule.

AFP is a glycoprotein (contains 4.5% carbohydrates) consisting of 590 amino acids in human and has a molecular mass of 69–70 kDa. It has a V-shaped form, built of three domains, which is subject to conformational changes under the influence of changes in the external environment conditions, such as fluctuations in pH, osmolality, temperature, etc. It belongs to the albuminoid gene family, which includes albumin (ALB), vitamin D binding protein (DBP), alpha-fetoprotein (AFP), alpha-albumin (alpha-ALB) and the AFP-related gene (ARG) protein. It acts as a plasma carrier for many different ligands, such as fatty acids, retinoids, steroids, bilirubin, flavonoids, phytoestrogens, heavy metals, dyes, dioxins and various drugs. It can bind to many different types of membrane receptors and intracytoplasmic proteins (including numerous as yet undescribed) and block or enhance responses from intracellular signaling pathways. Because of this, it has the ability to modulate cell growth in fetal life and during tumorigenesis (in both situations, it can be an enhancer or an inhibitor) [2,4,5,7].

## 2. Role of AFP in Carcinogenesis Process

We have distinguished two basic forms of alpha-fetoprotein: the nAFP (native AFP) plasma protein in fetal circulation and the tAFP (tumor AFP), sourced from hepatocellular carcinoma (HCC). As far as AFP has proven to be a valuable ally for the fetus, in HCC patients it turns into a treacherous foe [8,9]. nAFP in the fetal stage is responsible for stimulating cell and tissue growth, with an evident benefit for the fetus. Adult people record trace concentrations of nAFP (of 5–8 ng/mL), which mainly conditions physiological cell regeneration and hematopoiesis. The tAFP is not a mutated form of nAFP; it differs only slightly in structure (in the glycosylation process) [10,11]. In the case of the development of hepatocellular carcinoma, the growing tAFP level secures the supply of the required growth factors and nutrients for the forming neoplasm, creates and supports tumor-growth inductive conditions of immunosuppression, and causes drug resistance and metastases. To sum this up, it is responsible for features of malignancy [4,5].

tAFP supports carcinogenesis by the multidirectional impact it has on cellular signaling paths in hepatocytes, resulting in stimulation of proliferation and growth of cancerous cells, blocking of apoptosis, increasing drug resistance, and initiation of metastatic processes. The higher the tAFP level, the more severe these processes, and the worse the prognosis for the patient. tAFP binds to its membrane receptor, and then with cytoplasm proteins, increases the concentration of cAMP (cyclic AMP) and calcium ions in the cytoplasm. What follows is the activation of the cAMP-PKA cellular pathway, which increases the promotion of c-jun, c-fos, RAS oncogene expression and facilitates the transition from G1 to S phase in the cellular cycle and the stimulation of angiogenesis (proliferation) [8,12]. By binding with caspase 3, it blocks the cellular signal pathway from caspase 8 to 3 (blocking apoptosis) [13]. By binding with the ATRA (all-trans retinol acid) receptor, that is, the RAR-beta (beta-retinol acid receptor), it inhibits its transport to the nucleus, binding with HRE (hormone response element) within the DNA and synthesis of GADD153/CHOP/DDIT3 (growth arrest and DNA damage-inducible protein 153/C/EBP homologous protein/DNA damage-inducible transcript 3), blocking apoptosis [6,14]. By binding with PTEN (phosphatase and tensin homologue) [15], it activates the PI3K/P-AKT/mTOR cellular pathway, disabling the autophagy capacity of liver cancer cells, stimulating cancer cell growth and drug resistance [5,8,10,16]. The same pathway is used to excite the expression of 4 proteins responsible for metastatic processes: K19 (keratin-19), EpCAM (epithelial cell adhesion molecule), MMP 2 and 9 (matrix metalloproteinases 2 and 9) and CXCR4 (CXC chemokine receptor 4) [8,12,13,17]. Summary of the pathways are presented in Figure 1.

## 3. AFP as an Oncological Carrier Protein for Drugs

Modern cancer treatment should be based on the delivery of drugs directly to cancer cells (targeted therapy). This increases their effectiveness and reduces toxicity to the surrounding healthy tissue. One of the unique features of AFP is its ability to deliver drugs that activate or inhibit growth directly to myeloid suppressor cells (MDSCs) of both regenerating and cancerous cells. The combination of AFP with toxins destroys cancer cells (chemotherapy) and stimulates the immune response of T and NK: natural killer lymphocytes (immunotherapy) [10]. The ability of AFP to bind toxins results from its specific spatial structure. AFP is V-shaped and has three specific peptide domains. Between domain I and III, there is a hydrophobic pocket that binds covalently hydrophobic ligands, e.g., drugs and toxins (see Figure 2). A potential issue with the clinical use of AFP is the dominance of albumins in the serum of adults due to competition in molecule transport. Immediately after contact, the serum-present albumins take over molecules from AFP from other nonvalent binding domains. However, this does not apply to substances covalently bound in the hydrophobic domain. In an environment of alkaline blood pH, AFP molecule conformation changes, protecting the transported molecule from being taken over by albumins. When AFP with the drug reaches the cell, in conditions of acidic pH inside the cell, conformational changes occur, and the drug is released. Unlike albumins, AFP is also more selective for target cell types (it transports only to cells with RECAF - AFP receptors) and binds certain types of drugs/toxins more strongly [5,18,19,20]. Since the 1990s, work has been underway to obtain a pharmacological form of AFP that is as simple as possible in synthesis, storage and method of its administration to the patient. So far, several forms have been considered as carrier protein for drugs and are presented below.

### 3.1. AFP—Whole Molecule (FL-AFP)

This might seem to be the perfect carrier protein. Its advantages are its simple design, biodegradability and lack of immunogenicity. FL-AFP can repeatedly transport molecules intracellularly (as opposed to other carrier proteins, such as monoclonal antibodies, peptides or other synthetic ones). When combined with a specific receptor, it gives back the molecule in the endocytosis process and returns to the bloodstream intact. However, from common clinical use, it is disqualified by the following disadvantages: large particle size, complicated production process, difficulties in administration, long-term storage, and the possibility of sudden and unpredictable activation of numerous additional biological functions of AFP. As a result of the exposure of the AFP molecule to variable conditions of the biochemical environment inside the body (fluctuations in pH, osmolality, temperature, etc.), its conformational changes are forced and an unexpected activation of additional functions may occur, e.g., stimulation of cell growth, stimulation of neovascularization, cell proliferation, etc. [21]. The result of this may be an unpredictable cascade of reactions with resulting harm to the patient, e.g., induction or intensification of carcinogenesis as a result of the effect on small, previously undiagnosed tumor foci [5,22,23]. Due to the above-described threat, the use of FL-AFP or rhAFP (recombinant form) for the treatment of patients is discouraged, and the peptides obtained from AFP are currently considered to offer a much safer form of therapy. In clinical trials to date, the following two (GIP-34 and GIP-8) are the most promising.

### 3.2. GIP-34 (Growth Inhibitor Peptide)

This was discovered by Gerald Mizejewski (USA) in 1993. This peptide is located deep in the third domain of native AFP [21]. It is composed of 34 amino acids and is similar in its structure to the family of heat shock proteins (HSP) [21,24]. It consists of three adjacent fragments: GIP 12, GIP-14 and GIP-8. It can be most easily isolated in the laboratory from transformed AFP, formed from nAFP, in a situation of stress and cellular shock (e.g., changes in temperature, pH, osmolarity), when conformational changes occur that expose peptides. The transformed AFP is designed to block cell proliferation processes until the external environment is stabilized. This is likely to avoid malformations and neoplasms in the fetus, but in adults, due to trace concentrations of nAFP, such a function is no longer possible [25]. The way GIP works depends on its concentration. It can act as a cell-penetrating peptide (CPP) or a calcium channel blocker. Cancer cells and cells intended for apoptosis have a characteristic negative electrical charge on the surface of their cell membranes (resulting from the characteristic phospholipid system), unlike healthy “positive” cells, which allows them to be selected for CPP action. CPP interrupts or disrupts the structure of the lipid bilayer of cell membranes and penetrates to the center of the cell, where it activates apoptosis. The primary role of GIP is to inhibit cell growth in fetuses and in cancerous tissue, but not in healthy, highly diverse tissue in adults. GIP is cytostatic, not cytotoxic. It blocks metastasis by inhibiting the adhesion of cancer cells to the extracellular matrix (ECM) of proteins such as collagen, fibronectin, laminin and blocking 90–95% of all stages of the platelet aggregation process and 95% of the angiogenesis process [18,21] Mass production of GIP would allow the spectrum of cancer drugs, for treatment both estrogen-dependent and independent cancers, to be expanded. GIP is effective in animal studies in nine types of cancers, including in MCF-7 estrogen-dependent breast cancer (50% to 80% reduction in growth) and slightly less in MDA-MB-231 estrogen-independent breast cancer (where it reduces its growth by 40%) [21,25,26]. The addition of GIP to chemotherapy/hormone therapy reduces resistance to tamoxifen in the treatment of breast cancer, reduces the risk of endometrial hypertrophy and endometrial cancer and the risk of venous thromboembolism. It also reduces the risk of resistance to herceptin and the risk of cardiac arrhythmias in this type of therapy. GIP eliminated the toxicity of doxorubicin administered to MCF-7 breast cancer cells in clinical trials [5]. It is active against prostate cancer cells (reducing the growth of their cell lines from 40% to 90%) and renal cancer (by 70%) [26]. Attached to radioiodine and technetium-99, it is used in the diagnosis of breast cancer [25]. In oncology, it can be used as a cancer cell sensitizer for the purpose of radio and chemotherapy [25]. The combination of GIP with heavy metals such as Fe, Cu, Co, Zn is tested in order to be applied in MRI and CT technology [25]. GIP is atoxic, with no unwanted side effects in use. Unfortunately, mass production of peptide-based drugs is limited due to the high costs of its synthesis and manufacture (GIP-34 more expensive than GIP-8) and the short half-life of these drugs due to degradation by endogenous peptidases [25].

### 3.3. AFPep (Cyclic Version of GIP-8)

Discovered by G. Mizejewski in 1993 during the enzymatic decomposition of GIP-34 [11]. It is a small cyclic peptide composed of 9 Glu-Lys-Thr-O-Val-Asn-O-Gly-Asn amino-acids, where O stands for hydroxyproline. At the cellular level, it acts as a multi-inhibitor of estrogen receptor phosphorylation kinases (FAK - focal adhesion kinase, c-kit: tyrosine-protein kinase KIT) after binding the ligand. It blocks the activation of the receptor and what is more, the effect of estrogens on cellular growth stimulation pathways (it does not destroy them or alter the metabolism) [27,28,29,30]. This potential has been exploited and tested for AFPep as a cell growth inhibitor in estrogen-dependent tumors. The main target is breast cancer, as 50–55% of all breast cancers are estrogen-dependent. In clinical studies in animals, AFPep combats xenografts of MCF-7 breast cancer and reduces ascites associated with advanced stages of breast cancer [13,18,31,32]. AFPep may potentiate the antiestrogenic effect of tamoxifen in the treatment of breast cancer (particularly in cases of increasing resistance), eliminating its effect on the endometrium (blocking endometrial cell hypertrophy) [27,28,29,33] and acting chemopreventively in breast cancer, especially in patients from risk groups [27,34]. AFPep, on the other hand, lacks negative immunosuppressive and carcinogenesis stimulating properties in the liver. It is also atoxic and has no side effects (it breaks down into simple amino acids). In animal trials, it did not affect the menstrual cycle or reproductive potential [27,35]. It can be administered parenterally or orally. The oral form is unique because, under normal conditions, gastric acid and digestive enzymes break down peptides, but the cyclic form is less sensitive to it. Unfortunately, the oral form has lower bioavailability and is characterized by greater variability in the measured plasma concentrations of treated patients, but these concentrations are nevertheless sufficient to destroy tumors [25,30,36,37].

### 3.4. AFP-Inhibiting Fragments (AIFs)

Classically, AIFs consist of peptides and fragments sourced from the III AFP domain. The peptide is usually a GIP or its analogues that has no affinity to AFPR. The AIFs include one of the following: AFP-3BC, rAFP3D or r3dAFP, which may transport drugs and be subject to endocytosis by tumor cells with high AFPR expression [10,38]. One of the roles envisaged for AIFs is the selective transport of cytotoxic drugs exclusively to cancer cells. In addition to the function of carriers, there have been attempts to design special AIFs to prevent AFP from combining with cytoplasmic proteins, which would block the activation of the cellular pathways responsible for the growth of cancer cells. For example, blocking the binding with caspase 3 would stimulate apoptosis, and by inhibiting the PTEN function and blocking the PI3K/AKT signal pathway, it would block cellular pathways, leading to multidrug resistance (MDR) [10].

## 4. RECAF (Alpha-Fetoprotein Receptor)

This is a membrane receptor for FL-AFP, but it does not have the ability to bind GIP34 and GIP8 [26]. It was isolated from the cell membranes of MCF-7 breast cancer [26]. It belongs to the family of scavenger receptors (SR) [39]. Its structure is not yet fully understood—there are probably two major membrane fractions and two soluble cytoplasmic fractions. It has the characteristics of a G protein–coupled receptor (GPCR), so the transmission of intracellular signals is consistent with the principles for this receptor (after attaching the ligand, the conformation of the intramembrane part of the receptor changes, which results in the activation of the G protein and further transmission of the signal to the inside of the cell) [5,10,40]. AFP, after binding to RECAF, is endocytosed and packaged together with the receptor in endosomal follicles and transported by the Golgi apparatus to cellular organelles, where it is degraded or initiates/blocks cellular signal pathways. RECAFs are mainly responsible for endocytosis, but they can also serve as scavengers or receptors for MDSCs chemotaxis [5]. They can be found on embryonic cells (but not on their adult counterparts), cancer cells of breast cancer, colorectal cancer, lung cancer, rectal cancer, non-Hodgkin lymphoma; MDSCs (myeloid-derived suppressor cells), multiplying hepatocytes and in high concentrations in plasma of cancer patients [10,38,41]. Laboratory studies demonstrate that RECAF can be used alone as an aspecific tumor marker, because, unlike AFP, it is found on the cell membranes of most types of tumors with a low degree of differentiation (and therefore, it cannot be said to be a universal type of marker) [39,42]. It has the potential to be used for screening, monitoring of treatment and relapses [41]. For example, the addition of RECAF to Ca-125 increases sensitivity and specificity in the diagnosis of ovarian cancer [23]. Exploiting the practical side of the AFP receptor leads to the creation of the drug called AIMPILA, which is an oral anticancer drug directed against RECAF. In its design, it is a noncovalent AFP complex (carrier) with atractisolide glycoside (inducing apoptosis) [38,40].

## 5. The Role of AFP in Cancer Immunotherapy

### 5.1. AFP and Immunotolerance in Hepatocellular Carcinoma

Hepatocellular carcinoma is the most common liver cancer and the third most common cause of cancer mortality worldwide [43]. Some 600,000 cases per year are diagnosed, mainly in Asia (50% of cases [44]) due to the high prevalence of HBV and HCV infections [45]. The therapy is mainly based on surgical treatment, thermo- and radio-ablation, chemoembolization, treatment with biological drugs and ultimately liver transplantation [45]. Since conventional chemotherapy does not work in the treatment of HCC (due to response rate of 0–25%) [6,46], over the last 20 years, therapeutic hopes have been connected with the developments of immunotherapy. Immunotherapy is not only supposed to destroy tumor cells, but also create lymphocytes with immune memory, which will prevent recurrence [45,46,47,48]. More than 50–70% of low-differentiated HCCs express tAFP [5,6,11,46]. tAFP produced by liver cancer has become the target of immunotherapy, as it is not only a marker of cancer but is also responsible for the immunosuppressive environment around the tumor that is conductive to the carcinogenesis [45,49,50,51]. AFP leads to inhibition of the immune response without causing a general impairment of immunity [52,53,54]. The primary cause of cancer immunosuppression is the formation of Tregs (T cell regulators) and the existence of MDSCs [45,54,55,56,57,58].

Formation of Tregs is suspected as the source of the lack of immune system response in the situation of tumor development and maternal–fetal tolerance during pregnancy [2,52,54]. The immunosuppressive effect of AFP in pregnancy results from the effect on the shift in the production of ThCD4 + lymphocytes towards Tregs in pregnant patients, which then circulate consecutively in the blood and settle there, increasing the infiltration of the uterus, and thus the response of ThCD8+ lymphocytes is blocked [34,54]. The formation of Tregs requires reduction of CD25 and CD28 activation markers on Th lymphocytes, with unchanged amount of CD71 (proliferation marker) [54]. Reduced production or impairment of Tregs may affect miscarriage (increased activation of ThCD8+ lymphocytes and decreased uterine infiltration in studies on mice) [54,59]. The formation of Tregs is a cascade of events, initiated by blocking the transformation of monocytes into fully functional dendritic cells (MDDC—monocyte derived dendritic cells) under the influence of tAFP. The impact on MDDC is long-lasting (it does not disappear even after the decrease in tAFP) [11,60,61]. In their place, the tolerogenic DC are formed. The primary role of dendritic cells is the presentation of foreign antigen (APC—antigen presenting cell) by MHC I and II pathways to ThCD4+ and CD8+ lymphocytes [11], which happens after combining CD80/86 with CD28 on the surface of APC [42] tAFP causes that on dendritic cells to lower the expression of HLA-DR, CD40, CD80, CD83, CD86, CD206, MHCII and increase the expression of CD14 [61,62]. It also leads to the lack of production/change in the structure of TLR4 (Toll-like receptor 4) on the surface of DCs, which are the receptors of lipopolysaccharides (LPS) of Gram-negative bacteria, and consequently to the blockade of the process of production of proinflammatory cytokines. Synthesis of Il-12 and TNF-alpha (responsible for stimulation of CD4+ lymphocytes and cytotoxic CD8+ lymphocytes) is reduced, and synthesis of anti-inflammatory IL-10 and TGF-beta increased [11,62]. tAFP causes differentiation of ThCD4+ lymphocytes into regulatory T cells through altered tolerogenic DCs [11,60,62]. In addition, tAFP causes apoptosis of NK cells or blocks their activation by dendritic cells, reducing the expression of CD1 (especially CD1d) on their surface [11,57,63],. This is important, since NK identify lipid antigens presented to them by dendritic cells using CD1d [61] Figure 3.

MDSCs are a heterogeneous mixture of immature myeloid cells that originate from myeloid hematopoiesis stem cells. They have CD11b, CD14, CD15 and CD33 antigens on their surface, a feature that they share with other myeloid cells [52,54]. They do not differentiate into macrophages, dendritic cells or granulocytes. MDSCs have an AFP receptor, and after its bonding, they block the activity of NK cells (an element of innate immunity) and T lymphocytes (blocking acquired immunity). Their activation is responsible for the reduction of inflammation (immunosuppression), which allows autoimmune diseases to be extinguished, but also constitutes the basis of immunotolerance in cancers. On the other hand, the destruction of MDSC can stimulate the immune system and lead to the destruction of cancer cells [55,56]. For this reason, the target of immunotherapy are the MDSCs and their destruction/destabilization, which improves the response of the immune system, activates NK and T cells, and stimulates the recognition and removal of cancer cells. Cancer immunotherapy is even better than classical immunotherapy, as it stimulates NK cells, which destroy cancer stem cells and metastases, but also reduces the number of Tregs [5,11] Figure 4.

### 5.2. AFP as a Vaccine

In clinical trials, cyclical administration in AFP (nAFP or rhAFP) and AFPR vaccines causes dysfunction of MDSCs, intensification of immune response, tumor destruction and prolongation of life [5,10,52,53]. rhAFP (recombinant human AFP) is similar in function to nAFP derived from natural sources, including the same immunosuppressive properties. Biological activities of AFP are dependent only on the presence of specific peptide groups [5]. Not a single known function of AFP is dependent on the glycosylation process and there are no differences in detection of either (nAFP and rhAFP) by commercially available enzyme immunoassays [47,48,49]. Large-scale use of nAFP is debated from an ethical point of view, as it can only be obtained from human fetal plasma. The rhAFP on the other hand is synthesized in the laboratory using E. coli bacteria/Saccharomyces cerevisiae yeast, but issues such as the formation of intrusions during production or issues with folding (and vice versa) of this complex spatial structure are described. In addition, solutions with higher concentrations are unstable, which is why there is no recombinant AFP on the market so far (this also includes AFPR) [5,64]. Generally, the immune response to pure AFP is poor (CD8+ lymph stimulation is observed, but no anti-cancer response [50,51]) and requires additional stimulation, such as a combination of AFP with daunorubicin. Daunorubicin itself destroys G-MDSC (granulocytic myeloid-derived suppressor cells) well, but once combined with AFP it also destroys M-MDSC (monocytic myeloid-derived suppressor cells), and these are more important in cancer immunosuppression than G-MDSCs [56,65].

### 5.3. TCR-T (T Cell Receptor–Transduced T Cells)

Like CAR-T (chimeric antigen receptor T cells), the therapy is based on modifying T cells to provide them with special receptors. While CAR-T attacks the natural antigens on the surface of the tumor, TCR-T only attack the antigens connected with MHC. T cells activate after the binding of TCR (specific T cell receptor) to the MHC/peptide complex [51,66]. The more specific the T lymphocytes are, the greater the chance of success of the therapy [44,57]. Autologous HCC—specific T lymphocytes are difficult to obtain from the patient and have a low affinity to HCC cells [51]. In the laboratory, TCR-T can only be obtained by genetic modification of the TCR receptor. HCC, unlike other solid tumors, does not reduce HLA expression to avoid immune response, so the HLA/AFP complex is readily available [44]. TCR for CD8+ against MHC/AFP (HLA-A2/AFP158–166) was obtained, and clinical trials are ongoing [11,36,51,66,67]. TCR is currently being sought for other HLA alleles (especially those popular in Asia) [10]. The problem with its popularization rests in the high toxicity of TCR-T in vitro therapy, up to and including the patient’s death. This is primarily due to on-target/off-tumor toxicity—the possibility of T cells attacking healthy tissue demonstrating low expression of tumor antigen. Secondly, it also includes the chaotic recognition of epitopes of normal proteins as affected ones. Thirdly, toxicity to its own proteins is presented by different HLAs [66,68,69].

### 5.4. Dendritic Cell Vaccine

The hope in the treatment of HCC is brought about by the vaccine from dendritic cells sensitized to AFP, which is to trigger a response from CD8+ cytotoxic T cells (CTL) and the destruction of HCC cells [6]. However, the main issue in HCC immunotherapy is the heterogeneity of tumor cells in the expression of the AFP gene and the presence of AFP antigens on the surface of hepatocytes. This is due to the genetic hepatocyte heterogeneity, the different degree of methylation of the AFP gene and regulation of its transcription process [43]. Immunotherapy using DCs is tested in lung, breast, melanoma, bile duct, prostate, kidney and gastrointestinal cancers [70,71]. Live dendritic cell vaccines are expensive and difficult to manufacture, quality control and store [36,71]. It is difficult to multiply such quantities of DCs to reach clinically usable amounts, and the ability to present antigen in DC varies depending on the method of administration of the tumor antigen [71]. The primary vaccine of HCC dendritic cells alone has been proven to be ineffective, so there are now attempts to improve it to enhance the response [36,72]. Activated CD4+ lymphocytes have the CD40L antigen on their surface, which can activate dendritic cells through interaction with CD40, and these in turn activate the CD8+ cytotoxic lymphocytes. Coupling dendritic cells with CD40L improves the effect of the HCC vaccine, increases the infiltration of cytotoxic lymphocytes and dendritic cells and apoptosis [72,73]. Another method is the addition of zoledronic acid to DC stimulation, which results in an increased release of interferon by CD8+ [45]. It is possible to culture DC with Il-2 and GM-CSF, which stimulates increased secretion of IL9, IL-15 and TNF [45,74].

### 5.5. ICIs (Immune Checkpoint Inhibitors)

PD-1 (programmed cell death protein 1) is a receptor expressed on T, B lymphocytes and macrophages. Once combined with PD-L1 (programmed death-ligand 1), it inhibits the activation of the immune system. AFP plays a major role in the stimulation of PD-L1 expression on HCC cells through increased HIF-1-alpha synthesis (stimulation of the PI3K/AKT cellular pathway) [10,75,76]. ICIs block the binding of PD-L1 to PD-1 on immune cells by stimulating the immune system to fight cancer, e.g., as is the case of PD-1 inhibitors (athezolizumab/nivolumab) and PD-L1 inhibitors (pembrolizumab/durvalumab) [77,78]. ICIs are combined with AIFs to enhance the effect (AIFs will reduce MDC activity and inhibitors stimulate T and NK lymphocytes) [10]. Unfortunately, the response rate to ICIs treatment is just 20% and is accompanied by a lot of adverse effects [44,75,77,79,80].

## 6. The Future of AFP in Oncology

Dynamically developing oncology treatment techniques are focused on highly precise treatment of cancer, with a reduction in risk to surrounding tissue and the risk of side effects. In the treatment of HCC as well as other cancers, AFP and its receptor can act as both a target and a drug. This prompts the search for new methods of immunotherapy, gene therapy, and analysis of the usefulness of substances sourced from the sphere of natural medicine.

### 6.1. Knockdown of the AFP Gene and AFP-siRNA

Knockdown of the AFP gene results in blocking the growth of HCC cells. AFP has a stimulating effect on the regulation of the cell cycle of the liver tumor [21], the suspension of its synthesis stops the cell cycle by delaying the transition from G1 to S. Silencing the AFP gene allows the synthesis of TNF-beta and the mutated p53 protein in HepG2 cells to be reduced, which prevents the apoptosis of tumor cells via the p53/Bax/caspase 3 pathway [6]. The normal p53 protein induces apoptosis by increasing the Bax/Bcl2 ratio. Silencing AFP also results in increased caspase 3 production and increased cell susceptibility to TRAIL (tumor necrosis factor–related apoptosis-inducing ligand), thus restoring the ability of caspase 3 to transmit signal and control cell apoptosis. Knockdown of AFP, together with treatment with ATRA (all-trans retinoic acid) and TRAIL, improves the effect of chemotherapeutic agents [30].

AFP-siRNA (small interfering RNA), considered the future of HCC treatment [36], was designed to suppress AFP genes in HCC tissue. Silencing the expression of AFP induces apoptosis of HepG2 hepatic cancer cells. siRNA is a double-stranded RNA molecule that silences the expression of genes with a homologous sequence. siRNA binds to the RISC (RNA-induced silencing complex) protein with ribonuclease activity, and then to the target mRNA molecule and cuts it into pieces. siRNA is effective in reducing both mRNA and protein levels [6]. siRNA is combined with nanoparticles that serve as a carrier for this drug and other therapeutic substances, protecting them from degradation and excretion, and increasing intracellular transport and bioavailability. Nanoparticles are preferentially accumulated in tumor tissue. Thanks to this combination, this treatment option is extremely specific and works only on liver cancer cells [57].

### 6.2. AFPmAb-PLGA-rhDCN

This is a type of gene therapy in the form of monoclonal antibodies against AFP, combined with a copolymer of lactic and glycolic acids (PLGA (poly lactic-co-glycolic acid)) and a recombinant human decorin gene. Decorins (a member of the proteoglycan protein family) are an inhibitor of proliferation, metastasis and angiogenesis in tumor tissue, capable of reversing drug resistance. They induce apoptosis of Hep2G cells by affecting the expression of caspase 3 genes (stimulation and Bcl-2 (blocking)) [81].

### 6.3. CAR-T (Chimeric Antigen Receptor T Cells)

A type of therapy based on such genetic modification of T cells (equipped with chimeric antigen receptors) to attack natural antigens on the surface of cancer cells (TCR-T attacks only the MHC-bound antigens). CAR-T from CD8+ T cells with AFP receptors is currently being investigated [5,53,82].

### 6.4. The Combination of rhAFP with Taxanes and 1′-S-1′-Acetoxychavicol Acetate (ACA)

The 1′-S-1′-acetoxychavicol acetate is obtained from the rhizomes of Alpinia conchigera, a plant of the ginger family found in Malaysia. ACA induces tumor cell apoptosis by blocking the pathway of NF-κB signals (used to control innate and acquired immunity) both in vivo and in vitro. rhAFP is the carrier for the ACA, which (thanks to its hydrophobic pocket) is poorly soluble in aqueous solutions. This enables the potential issue of poor ACA solubility in the aquatic environment to be avoided. ACA requires lower doses than chemotherapeutic agents, potentiates the effects of taxanes and reduces the side effects of cytotoxicity for healthy tissue. It is currently awaiting clinical trials [40,52,83].

### 6.5. EGCG (Epigallocatechin-3-Gallate)

This is the best-known polyphenol of green tea. It demonstrates anti-inflammatory, antioxidant, antiobesity and anticancer effects in many types of cancer. It stimulates apoptosis and autophagy in the HCC HepG2 cell line. The mechanism of its action remains unclear. It most likely acts by blocking different intracellular pathways. EGCG is likely to block AFP secretion from HCC cells due to reduced mitochondrial efficiency and intracellular transport disruption by impaired cytoskeletal function and leads to AFP intracellular aggregation, followed by autophagy of aggregates. Currently, special gold nanocages are being tested as EGCG carrier proteins, due to their low stability, bioavailability and rapid elimination [84,85,86].

## 7. Conclusions

AFP plays a crucial role both in the induction of growth and progression of HCC and is responsible for the immunosuppressive environment around this tumor.

The dual properties of AFP provide the possibility of future personalized chemotherapy (tumor destruction) and immunotherapy (reduced tolerance).

Only appropriately modified AFP peptide fragments (GIP-34, GIP-8, AIFs) can be used as medication in humans.

Certain forms of AFP, such as FL-AFP and rhAFP, should not be used in therapy as they may stimulate tumor growth.

The option of HCC therapy in the future may be the knockdown of the AFP gene, among others, by means of siRNA and a vaccine.

## Figures and Tables

**Figure 1 ijms-24-02539-f001:**
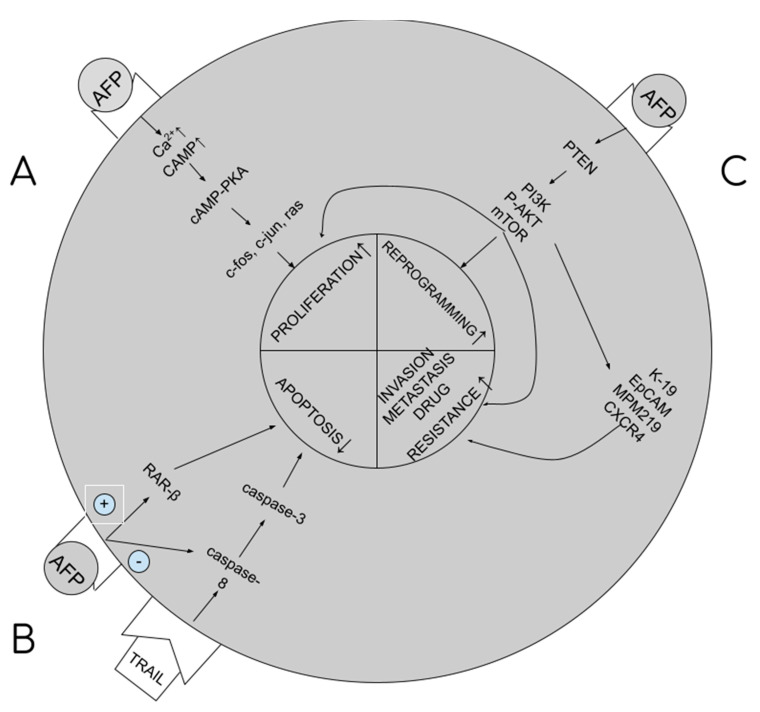
AFP influence on carcinogenesis. (**A**). Stimulation of proliferation. (**B**). Blockage of apoptosis. (**C**). Stimulation of reprogramming, metastasis and multidrug resistance development.

**Figure 2 ijms-24-02539-f002:**
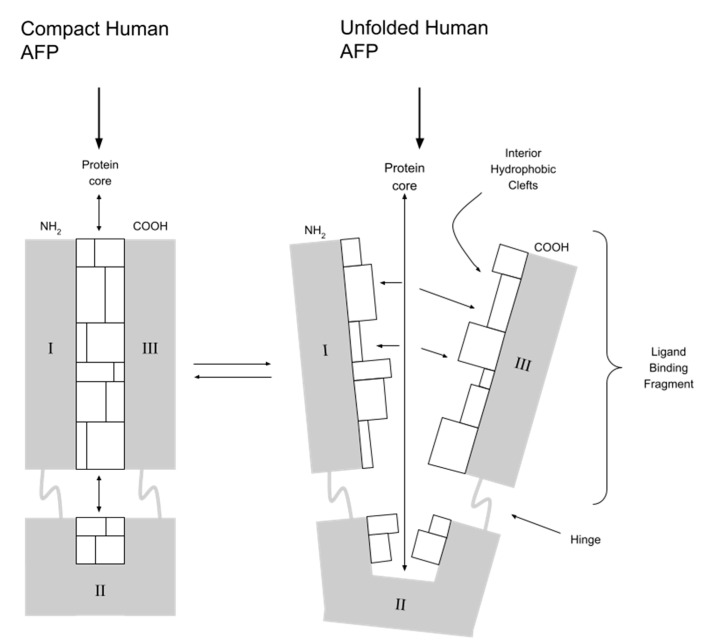
AFP and its hydrophobic pocket between domain **I** and **III**. **I**, **II**, **III**—peptide domains of human AFP.

**Figure 3 ijms-24-02539-f003:**
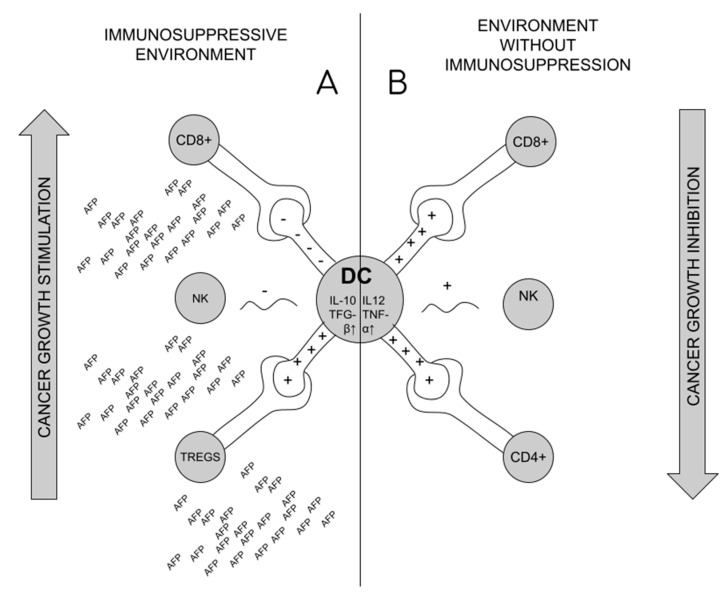
Dendritic cell role in cancer immunosuppression. (**A**). Under the influence of AFP, dendritic cells (DCs) stimulate the formation of Tregs, and block the activity of CD8+ and NK lymphocytes, which promotes the formation of an immunosuppressive environment and tumor development; (**B**). In the conditions of a healthy organism, DCs stimulate the activity of CD4+, Cd8+ and NK lymphocytes, which blocks the development of cancer.

**Figure 4 ijms-24-02539-f004:**
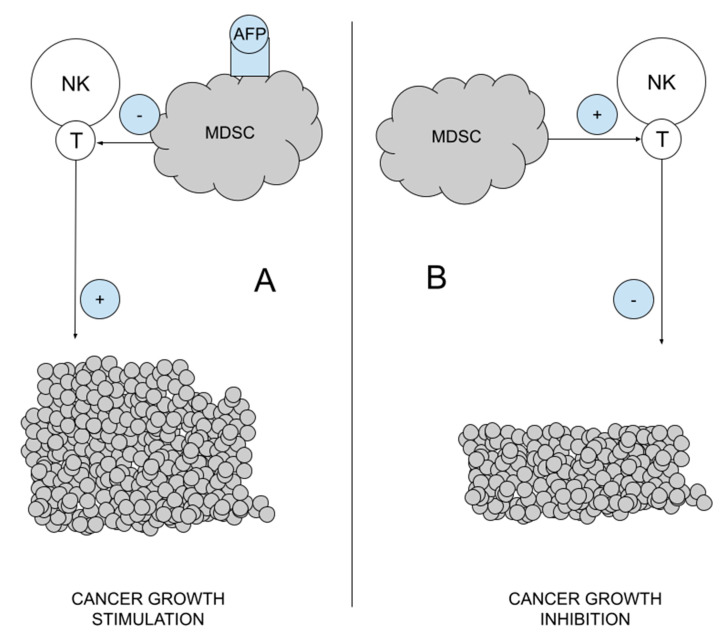
Myeloid-derived suppressor cells (MDSCs) role in cancer immunosuppression. (**A**). Under the influence of AFP, MDSCs block the activity of T and NK cells, which stimulates tumor development; (**B**). In a healthy organism, MDSCs activate T and NK lymphocytes, which blocks the development of cancer.

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
