# Peer review of "The Role of Alpha-Fetoprotein (AFP) in Contemporary Oncology: The Path from a Diagnostic Biomarker to an Anticancer Drug"

_ijms, 2023, doi:10.3390/ijms24032539_

Round 1

Reviewer 1 Report

The authors summarized the role of AFP and its multiple forms in oncologic diagnostics and treatment. In general the manuscript is well-written and easy to follow, and is a good fit for the journal. It would be better to have high-resolution figures and add a summary paragraph in each of the figure legend to better explain the figures. 

Author Response

Dear Editors,

Dear Reviewer 1,

This document includes our responses to the Reviewer’s comments on our manuscript “The role of alpha-fetoprotein (AFP) in contemporary oncology – the path from a diagnostic biomarker to an anticancer drug.” We appreciate the time and effort that the Reviewer dedicated to providing feedback on our manuscript and we are grateful for the insightful comments on and valuable improvements to our paper. We have incorporated all the suggestions made by the Reviewer. These changes are highlighted in yellow in the manuscript. Please see below for point-by-point responses to the Reviewer’s comments and concerns. All page numbers refer to the revised manuscript file with tracked changes. We trust that the manuscript in its present revised version will meet the criteria for publication.

Kindest regards,

Joanna Głowska-Ciemny, MD

Responses to Reviewer’s Comments

Reviewer 1

It would be better to have high-resolution figures and add a summary paragraph in each of the figure legend to better explain the figures. 

We sincerely appreciate the valuable comment. According to the Reviewer's suggestion, the figures and summary paragraphs have been changed as requested. We trust that now it is clearer.

Reviewer 2 Report

In the submitted review manuscript, the authors discuss the activity and role of alpha-fetoprotein in oncologic diagnostics and treatment. The review article focuses on the current and prospective therapies using AFP and its derivatives, and the gene therapy options along with the benefits and risks associated with use of AFP in oncologic therapy. Authors provide a brief introduction of the emergence of AFP and the biological activity of AFP in therapy.  The authors describe the role of AFP in carcinogenesis, in oncological drug transport, and in cancer immunotherapy. The perspective applications of AFP have also been discussed to provide an overall outlook of the potential biological activity as well as potential applications of AFP.

The manuscript is well written, concise, and focused to provide an overall view of past and current research regarding AFP as well as potential future use of AFP. The discussed topic is within the scope of the Journal and would be useful for readers of the research field, early-stage researchers in particular.

The manuscript is suitable for publication in the journal International Journal of Molecular Sciences, provided the authors address the following points:

·      The introduction section is very concise, I would suggest the authors revise the introduction section and make it more descriptive to provide a better insight into the importance and roles of AFP.  The benefits and the threat of AFP needs to be discussed in more detail along with the main biological activities of AFP in the literature, with suitable references.

·      Lack of references: The authors need to add more appropriate references in each section to make the introduction and other discussion sections more complete. Including, additional references would make it a much better read. This goes for the entire review manuscript. Please revise the list of references.

Author Response

Dear Editors,

Dear Reviewer 2,

This document includes our responses to the Reviewer’s comments on our manuscript “The role of alpha-fetoprotein (AFP) in contemporary oncology – the path from a diagnostic biomarker to an anticancer drug.” We appreciate the time and effort that the Reviewer dedicated to providing feedback on our manuscript and we are grateful for the insightful comments on and valuable improvements to our paper. We have incorporated all the suggestions made by the Reviewer. These changes are highlighted in yellow in the manuscript. Please see below for point-by-point responses to the Reviewer’s comments and concerns. All page numbers refer to the revised manuscript file with tracked changes. We trust that the manuscript in its present revised version will meet the criteria for publication.

Kindest regards,

Joanna Głowska-Ciemny, MD

Responses to Reviewer’s Comments

Reviewer 2

The introduction section is very concise, I would suggest the authors revise the introduction section and make it more descriptive to provide a better insight into the importance and roles of AFP.  The benefits and the threat of AFP needs to be discussed in more detail along with the main biological activities of AFP in the literature, with suitable references.

We sincerely appreciate the valuable comments. We have reviewed the relevant literature and rearranged the Introduction Section. It has been expanded and properly quoted. We trust that now it is clearer.

Lack of references: The authors need to add more appropriate references in each section to make the introduction and other discussion sections more complete. Including, additional references would make it a much better read. This goes for the entire review manuscript. Please revise the list of references.

Thank you very much for pointing this out. We have added appropriate references and expanded the list of them to 89 entries, adequately quoting them throughout the whole paper.

Reviewer 3 Report

In this manuscript by Joanna Głowska-Ciemny, et al., entitled "The role of AFP in contemporary oncology – the path from a diagnostic biomarker" showed the role of AFP in hepatocellular carcinoma and summarized AFP inhibitor and AFP-related molecules. I think it needs significant revision due to inadequate references cited and the figures are difficult to understand. My comments are listed below.

Major points,

1. line 71 and line 95: The authors say “drug transporter” in the text, but isn't transporter basically intermembrane transport and AFP is more appropriate terminology for cargo protein?

2. line 79: The authors mention the structure of AFP, but the lack of a diagram makes it difficult to understand. It should be illustrated.

3. line71-93: Only reference [4] is shown, which is insufficient..

4. line: Only reference [4] is shown, which is insufficient.

5. References [26] and [29] are the same.

6. Figure 2 and 3: I can't understand what you are trying to show. Please put figure legends on it.

7. line 282-292: The authors mention recombinant AFP, shouldn't you mention glycans here?

8.line 356: The authors use the term “inactivation”, but I think down regulation is appropriate for knockdown.

9: line 356 and 368: Since both contents are about knockdown, shouldn't they be combined into one part?

10: line 387: The authors discuss CAR-T, but the references cited are inadequate. They should cite the original study, not the review. (Clin. Cancer Res. 2017; 23: 478-488)

11. Overall, many parts of the report consist only of quotations from the review, so it would be better to cite the original work.

Minor points,

1. line 61: The authors present only GADD153, but isn't DDIT3 the official name and CHOP and GADD153 are synonyms?

2. line 62: Isn't 1533 an error for 153?

3. line 192: It says “GIP and AFPep”, but AFPep is GIP-8, so the notation must be wrong.

4. line 195: GPCR-G?

5. line 233: Shouldn't the abbreviation be on line 205?

Author Response

Dear Editors,

Dear Reviewer 3,

This document includes our responses to the Reviewer’s comments on our manuscript “The role of alpha-fetoprotein (AFP) in contemporary oncology – the path from a diagnostic biomarker to an anticancer drug.” We appreciate the time and effort that the Reviewer dedicated to providing feedback on our manuscript and we are grateful for the insightful comments on and valuable improvements to our paper. We have incorporated all the suggestions made by the Reviewer. These changes are highlighted in yellow in the manuscript. Please see below for point-by-point responses to the Reviewer’s comments and concerns. All page numbers refer to the revised manuscript file with tracked changes. We trust that the manuscript in its present revised version will meet the criteria for publication.

Kindest regards,

Joanna Głowska-Ciemny, MD

Responses to Reviewer’s Comments

Reviewer 3

Major points,

  1. line 71 and line 95: The authors say “drug transporter” in the text, but isn't transporter basically intermembrane transport and AFP is more appropriate terminology for cargo protein?

We sincerely appreciate the valuable comments. According to the reviewer's suggestion we have reviewed the relevant literature and agree that the term ”transporter” for AFP should be changed. We suggest the term ”carrier protein” instead of “cargo protein”. AFP (as a carrier) transports many substances in the bloodstream (as was mentioned in the Introduction Section) and is responsible for the transmembrane transport of the transported molecules inside the cell.  When it turns “empty” back to the bloodstream via Golgi apparatus/ secretory vesicles/ cell membrane as “cargo protein” it seems to be a short episode. Thus the term “carrier” in our opinion more precisely emphasizes the role of AFP. 

  1. line 79: The authors mention the structure of AFP, but the lack of a diagram makes it difficult to understand. It should be illustrated.

Thank you very much for pointing this out. We have added Figure 2 in the appropriate position in the revised manuscript.  We trust that now it is clearer.

  1. line71-93: Only reference [4] is shown, which is insufficient.

We sincerely appreciate the valuable comments. Reference list has been updated as suggested by the Reviewer and has been adequately quoted throughout the paper.

  1. line: Only reference [4] is shown, which is insufficient.

We sincerely appreciate the valuable comments. Reference list has been updated as suggested by the Reviewer. 

  1. References [26] and [29] are the same.

Thank you very much for pointing this out. Reference list has been reviewed and corrected as suggested by the Reviewer. 

  1. Figure 2 and 3: I can't understand what you are trying to show. Please put figure legends on it.

We sincerely appreciate the valuable comments. We have added notes to make these figures clearer in the revised manuscript. 

  1. line 282-292: The authors mention recombinant AFP, shouldn't you mention glycans here?

According to the Reviewer's suggestion the section “AFP as vaccine” has been expanded including information about the differences between nAFP and rhAFP. The significance of glycans in the structure, function/activities and detection of nAFP and rhAFP was described. We trust that now it is clearer.

8.line 356: The authors use the term “inactivation”, but I think down regulation is appropriate for knockdown.

Thank you very much for pointing this out.  The term “inactivation” has been changed into “knockdown” in the whole manuscript as suggested.

9: line 356 and 368: Since both contents are about knockdown, shouldn't they be combined into one part?

This section has been improved as suggested. We combined it into one part “ Knockdown of the AFP gene and AFP-siRNA”. We trust that now it is clearer. 

10: line 387: The authors discuss CAR-T, but the references cited are inadequate. They should cite the original study, not the review. (Clin. Cancer Res. 2017; 23: 478-488)

We sincerely appreciate the valuable comments. Reference list has been reviewed and corrected and adequately quoted in this section as suggested by the Reviewer.

  1. Overall, many parts of the report consist only of quotations from the review, so it would be better to cite the original work.

According to the Reviewer's suggestion, the reference list has been expanded in many valuable original works and adequately quoted.

Minor points,

  1. line 61: The authors present only GADD153, but isn't DDIT3 the official name and CHOP and GADD153 are synonyms?

We thank the Reviewer for the valuable comments and suggestions. We have mentioned the synonyms in line 76 as suggested. 

  1. line 62: Isn't 1533 an error for 153?

Thank you very much for pointing this out.  Typing errors have been avoided from all the sections. We apologize for this. 

  1. line 192: It says “GIP and AFPep”, but AFPep is GIP-8, so the notation must be wrong.

Thank you very much for pointing this out. These errors have been corrected. We meant GIP-34 and GIP-8 (AFPep). We apologize for this. 

  1. line 195: GPCR-G?

Thank you very much for pointing this out.  Typing errors have been avoided from all the sections. We apologize for this. 

  1. line 233: Shouldn't the abbreviation be on line 205?

Thank you very much for pointing this out.  This error has been corrected. We have added full words after the abbreviation at the first time this word appeared. We apologize for this.

Round 2

Reviewer 3 Report

Thank you for sending the revised manuscript. I think the reviewer's point has been answered adequately. I too would prefer the term "carrier protein" for AFP. This revised manuscript has been improved in a very understandable way and deserves to be published in the IJMS journal.

This is a very minor point, but I think it would be more accurate to add "in human" since the 590 residues in line 36 are about human AFP (for example, in mice, 588 residues).

Author Response

Dear Editors,

Dear Reviewer 3,

This document includes our response to the Reviewer’s comment on our manuscript “The role of alpha-fetoprotein (AFP) in contemporary oncology – the path from a diagnostic biomarker to an anticancer drug.” We want to thank reviewer for kind comment and useful suggestion that has been taken into the due account in the present MS revised form. This change is highlighted in green in the manuscript. Please see below for response to the Reviewer’s comment.

Kindest regards,

Joanna Głowska-Ciemny, MD

Responses to Reviewer’s Comments

Reviewer 3

“This is a very minor point, but I think it would be more accurate to add "in human" since the 590 residues in line 36 are about human AFP (for example, in mice, 588 residues).”

According to reviewer’s suggestion line 36 has been changed as requested. We trust that now is clearer.